# Home range and activity budget in the Falkland Steamer Duck (*Tachyeres brachypterus*)

**Alix M. I. Kristiansen**[1,2]*, **Alastair M.M. Baylis**[3], **Sébastien A.P. Dupray**[4], **Luc Lens**[2], **Heather Q. Mathews**[1], **Lucy J. Mitchell**[2], **John P. Y. Arnould**[1]

**1** School of Life and Environmental Sciences, Deakin University, Burwood, Victoria, Australia, **2** EcoBird, Ghent University, Ghent, Belgium, **3** South Atlantic Environmental Research Institute, Stanley, Falkland Islands; **4** Nelson Mandella University, Port Elizabeth, South Africa

* alix.kristiansen@gmail.com

## Abstract

The Falkland Islands support globally important populations of seabirds and coastal birds, underscoring their value for international conservation efforts. However, substantial knowledge gaps impede the development of coherent species management plans. This study focused on the endemic Falkland Steamer Duck, a territorial waterfowl only found around the archipelago, which has remained largely understudied and lacks fundamental ecological information critical to its conservation. To estimate home range sizes, habitat use, and activity budgets, we deployed GPS devices on 29 ducks from two locations (Bleaker Island and Stanley Harbour). Daily travel distances increased with proximity to ponds, kelp beds, and human infrastructures, within each duck's home range. Absolute area, but not proportion of, kelp significantly influenced home and core range. Patrolling males and incubating females spent less time travelling (respectively 31.4±2.5% and10.3±0.9%), with females spending the most time on land (70.0±2.4%) in line with their breeding role. Foraging time increased closer to kelp, and within larger areas of kelp, as well as closer to human infrastructure. Kelp beds are present in coastal waters all over the archipelago and, consequently, likely influence the distribution and density of Falkland Steamer Ducks. Therefore, any changes in their absolute area are expected to negatively impact the species. Preserving the kelp beds would therefore ensure a stronger resilience of the Falkland Steamer Duck in the context of ongoing climate change.

## Introduction

Understanding how much space a species needs, the habitat type it uses, and its intra- and inter-specific interactions is central to understanding its ecology. Estimation of home range, the geographic space which allows the individual to feed, mate and raise its young [1], is a fundamental tool for conservation and decision-making, and has been used for example, to design Marine Protected Areas [2,3] or to define Important Bird Areas [4,5]. These estimates are particularly crucial for islands, which

**Data availability statement:** The data are now available with the following DOI: https://doi.org/10.5281/zenodo.15309729.

**Funding:** LL and LM's work was partially supported by the Methusalem project (01M00221). Fieldwork was funded by the Shackleton Scholarship Fund (SSF22-019-ADC-Kristiansen and SSF23-021-ADC-Kristiansen) and the Falkland Environmental Study Budget (ESB122022). The funders had no role in the study design, data collection and analysis, decision to publish or preparation of the manuscript. There was no additional external funding received for this study.

**Competing interests:** There were no competing interests. This does not alter our adherence to PLOS ONE policies on sharing data and materials.

hold a higher concentration of often endemic species of limited distribution than the mainland [6]. Additional constraints to endemic species success, such as competition with invasive species and climate change [7,8], often result in data limitations hindering their conservation.

Understanding how a species' home range is shaped, and what resources and environmental factors define it, can facilitate better, more informed responses, should conditions, and consequently the species' population status, change [9]. For territorial species, their defended range is most likely a smaller component of their overall home range, and here they rely on resources that are of relatively constant value across the breeding period [10]. Territories of marine waterfowl encompass both marine and terrestrial habitats, reflecting their diverse needs. Most studied species are found in North America and Europe [11–13], and are migrants for which breeding and/or wintering grounds have been well described.

In contrast, relatively little is known of the marine waterfowl of South America and the Falkland Islands [14,15]. The Falkland Steamer Duck (*Tachyeres brachypterus*) is a member of the genus *Tachyeres,* found solely in South America and the Falkland Islands [16,17] (Fig 1). The genus comprises one flying species (*T. patachonicus* King, 1831) and three flightless species (*T. pteneres* (Forster, 1844), *T. leucocephalus* (Thompson, 1981), *T. brachypterus* (Latham, 1790)). Individuals of this genus form apparently long-term pair-bonds [18], guarding well-defended territories year-round [19], with incubation conducted solely by females [20] and the territory revolving around her [21]. The Falkland Steamer Duck, one of only two endemic bird and mammal species on the Falkland Islands, along with Cobb's wren *(Troglodytes cobbi)* , is ubiquitous along the coastlines. However, its current population size is unknown [22], and there is presently limited information on its movement or the factors that might influence their territory size [23]. Despite the lack of detailed information, Falkland Steamer Ducks are currently classified as "Least concern" by the IUCN [24].

Studies on Steamer ducks in South America thus far comprise information on both territory defence and nesting behaviour. In Argentina, the White-headed Steamer Ducks (*T. leucocephalus*) were found to select nest sites in areas with high proportions of shrub vegetation [25]. On the Falkland Islands, on the south bank of Stanley Harbour, one pair of Falkland Steamer Ducks had a nest around 0.8 km away from the shore, on the north side of the harbour [26], comprising a high proportion of shrub vegetation and ferns. The same study also reported two pairs nesting in tussac, suggesting similar nesting habitat as *T. leucocephalus*. Additionally, Falkland Steamer Ducks are believed to rely on access to freshwater [27], as a potential way to compensate for any salt stress linked to its diet [28].

The coastal ecosystem of the Falkland Islands presents a wasp-waist structure [29]. This type of ecosystem, instead of being either bottom-up or top-down driven, depends on its intermediate trophic level species [30]. A main feature of the Falkland Islands nearshore coastal ecosystem is the kelp beds, mainly of *Macrocystis pyrifera* [31]. This macroalgae provides several ecosystem services, hosts a diversity of species, dominated by Gastropods, Ascidiacea and Demospongia [32]. In a previous

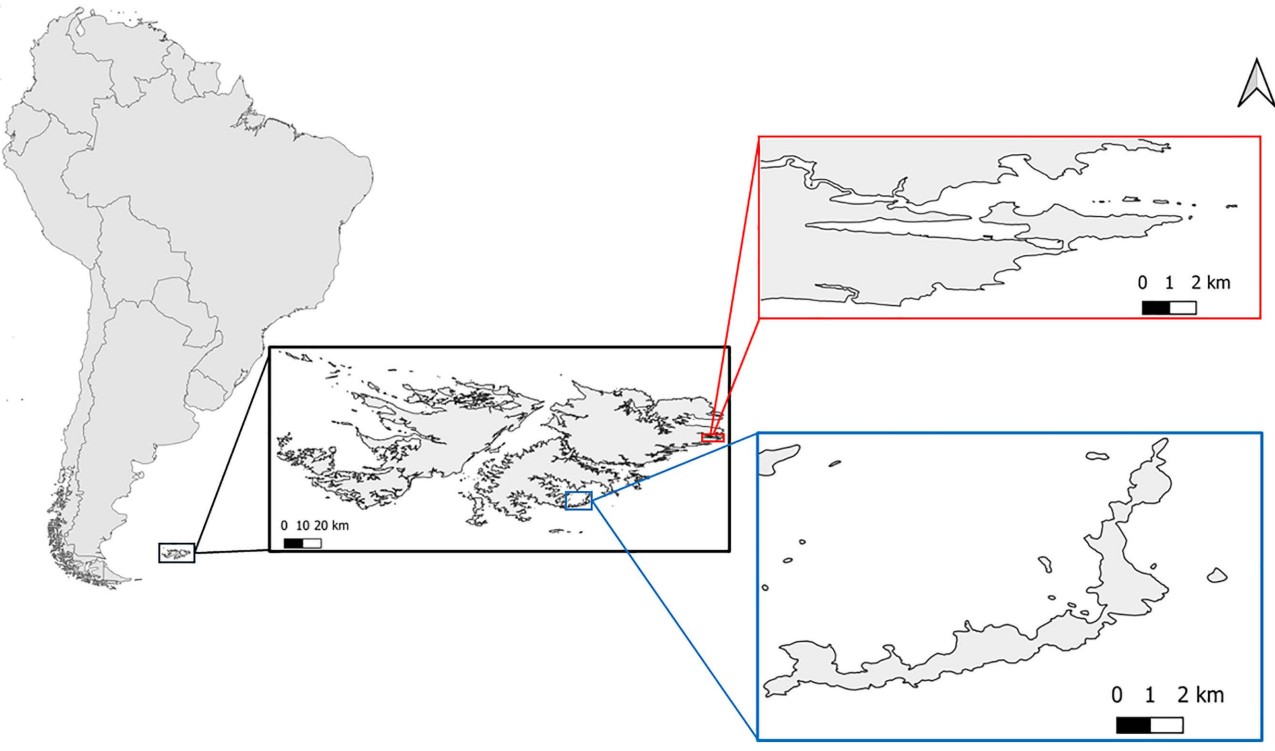

**Fig 1. Map showing the study site: Stanley Harbour (51°41'15"S 57°50'15"W, in red) and Bleaker Island (52°12'24"S 58°51'02"W in blue).** The coastline was represented by a shapefile provided by SAERI (FK-UKHO-414) and the South America polygon originates from rnaturalearth package.

Falkland Steamer Duck study, individuals were consistently observed less than 100 m away from surface-visible kelp beds, suggesting the importance of kelp beds to Falkland Steamer Ducks [27].

The scant, disparate pieces of information available on the Falkland Steamer Duck and its ecology leave many questions unanswered. Therefore, we sought to undertake a more detailed ecological tracking study to collect high resolution data on their movements and behaviour. Further, to identify key factors influencing Falkland Steamer Duck behaviour, along with collected GPS data, we recorded a number of variables that we believed to be environmentally relevant based on literature and field observations. Kelp beds and ponds were chosen as they are considered important resources for Falkland Steamer Ducks [19,26,27] and, thus, a good predictor of foraging habitat [18,19]. Human disturbance has been shown to affect anatid species [33] and was therefore also included in the analyses in the form of measured linear distances from bird GPS locations to settlements and roads. We hypothesised that (*i*) distance to kelp beds, ponds and human structures (*i.e.,* roads and settlements) influences daily activity budget (*i.e.,* distance travelled per day and proportion of time spent foraging); (*ii*) absolute area of kelp cover and proportional area of kelp beds constrains home and core range size; and (*iii*) breeding status, sex and study site impact time spent on land, home and core range size and daily activity budget (behaviour varies with breeding stage and habitat quality).

## Methods

### Study site and animal handling procedures

The study was conducted around Stanley Harbour and Bleaker Island, East Falkland (Fig 1). Stanley Harbour is a sheltered natural embayment which hosts the city of Stanley (human population: 2,974 – Fig 1). The terrestrial vegetation

differs between the urbanised areas (gardens, including introduced tree species) and the embayment coastline (mix of fern beds and shrub [34]). The coastline is comprised of stony beaches, with the exception of the sandy beaches found in York Bay and Surf Bay [15]. The density of Falkland Steamer Ducks was estimated to be of 7.7 pairs·km$^{-1}$ along the coastline of Stanley Harbour and its surroundings (Fig 1, [35]). In contrast, Bleaker Island has a more diverse coastline, with a topography which varies between cliffs, coves with pebbly beaches, and a long (1.6 km) sand beach on the east side of the island. There is a small settlement on the island that hosts a variable population of 5−20 people, depending on the number of tourists. The density of breeding pairs of Falkland Steamer Ducks in the Bleaker Island study area was estimated to be 9.8 breeding pairs·km$^{-1}$ of coastline (*Kristiansen et al in prep.*).

Animal handling procedures were conducted under the approval of the Falkland Islands Government (Research Licence R25.2022). Data collection occurred during the austral summer breeding periods of 2022/23 and 2023/24. Adult individuals were captured using a noose pole while roosting on the beach and placed in a cloth bag for weighing using a spring scale (Salter, Bristol, UK, 5.00 ± 0.25 kg). The sex of individuals was determined based on plumage characteristics [36]. Breeding status was extracted from a concurrent breeding phenology survey, which ran from September 2023 to February 2024 (*Kristiansen et al in prep.*). Breeding status was divided into four categories: incubating females; male partners of incubating females (hereafter, patrolling male); chick-rearing individuals (both sexes); non-breeding individuals (both sexes; 29) and assigned for the GPS tracking period.

A GPS data logger (IgotU GT120B (14.9 g), G2S (14.4 g) or G6S (20.9 g), MobileAction, Taiwan) sealed in heat-shrink plastic (60 x 40 x 11 mm) was then attached to dorsal feathers between the scapula using waterproof tape (Tesa Tape® 4651, Beiesdorf, AG, Germany (Wilson et al. 1997)). The GPS data loggers were programmed to record locations at 2 min intervals. Individuals were then released at the point of capture and observed remotely to ensure they resumed normal behaviours. Handling procedures lasted <30 min. Data were downloaded every 1–2 d from the device *in situ* using a Bluetooth® connection onto a mobile data storage unit. Data on the movements of free-ranging Falkland Steamer Ducks comprising >1 d of records were obtained from a total of 29 individuals (21 males, 8 females – Table 1, Fig 2), during 2 breeding seasons (austral summer 2022–2023 and austral summer 2023–2024). Tracking duration varied between 1 and 43 d (16.9 ± 1.0 d). Of the 8 females, 5 were observed to be incubating, 2 were chick-rearing and 1 was assumed to have failed breeding. Three of the males were observed to be chick-rearing, 4 were captured while patrolling their territory alone but seen with a partner at other times (and, thus, presumed to be partnered with an incubating female), and the remainder were individuals of a pair that were assumed to have failed breeding or did not engage in breeding. No two individuals tagged originated from the same breeding pair.

### Data analysis

All analyses were conducted in the R statistical environment (v. 4.4.2, R CoreTeam, 2018). Raw movement tracks were filtered to remove erroneous locations using a maximum travel speed of 38.6 km·h$^{-1}$, corresponding to steaming speed

**Table 1. Summary table of travelled distances along the trajectory of movement, core and home range depending on sex and breeding status. All values represent the mean ± standard deviation. \*: Home and core ranges were computed for 17 individuals: 3 females, 14 males; 8 non-breeding individuals, 6 chick-rearing individuals, 1 incubating female and 2 patrolling males.**

| | | Weight (kg) | Mean distance (km.d$^{-1}$) | Maximum distance (km.d$^{-1}$) | Core range (ha)* | Home range (ha)* |
|---|---|---|---|---|---|---|
| Sex | Female (n = 8) | 3.66 ± 0.18 | 8.63 ± 0.64 | 12.16 ± 1.06 | 1.18 ± 0.62 | 8.17 ± 4.97 |
| | Male (n = 21) | 4.68 ± 0.07 | 10.79 ± 0.61 | 14.03 ± 0.93 | 3.93 ± 1.08 | 21.76 ± 7.34 |
| Breeding status | Incubating female (n = 5) | 3.21 ± 0.010 | 8.52 ± 0.96 | 13.01 ± 1.49 | 2.42 | 17.97 |
| | Patrolling male (n = 4) | 4.78 ± 0.19 | 8.04 ± 0.04 | 12.22 ± 1.59 | 6.56 ± 1.65 | 39.17 ± 20.30 |
| | Chick rearing (n = 7) | 4.58 ± 0.22 | 10.35 ± 1.04 | 12.41 ± 1.41 | 2.55 ± 1.56 | 10.44 ± 6.01 |
| | Non-breeding (n = 13) | 4.78 ± 0.19 | 11.41 ± 0.75 | 14.70 ± 1.27 | 3.46 ± 1.51 | 21.28 ± 11.35 |

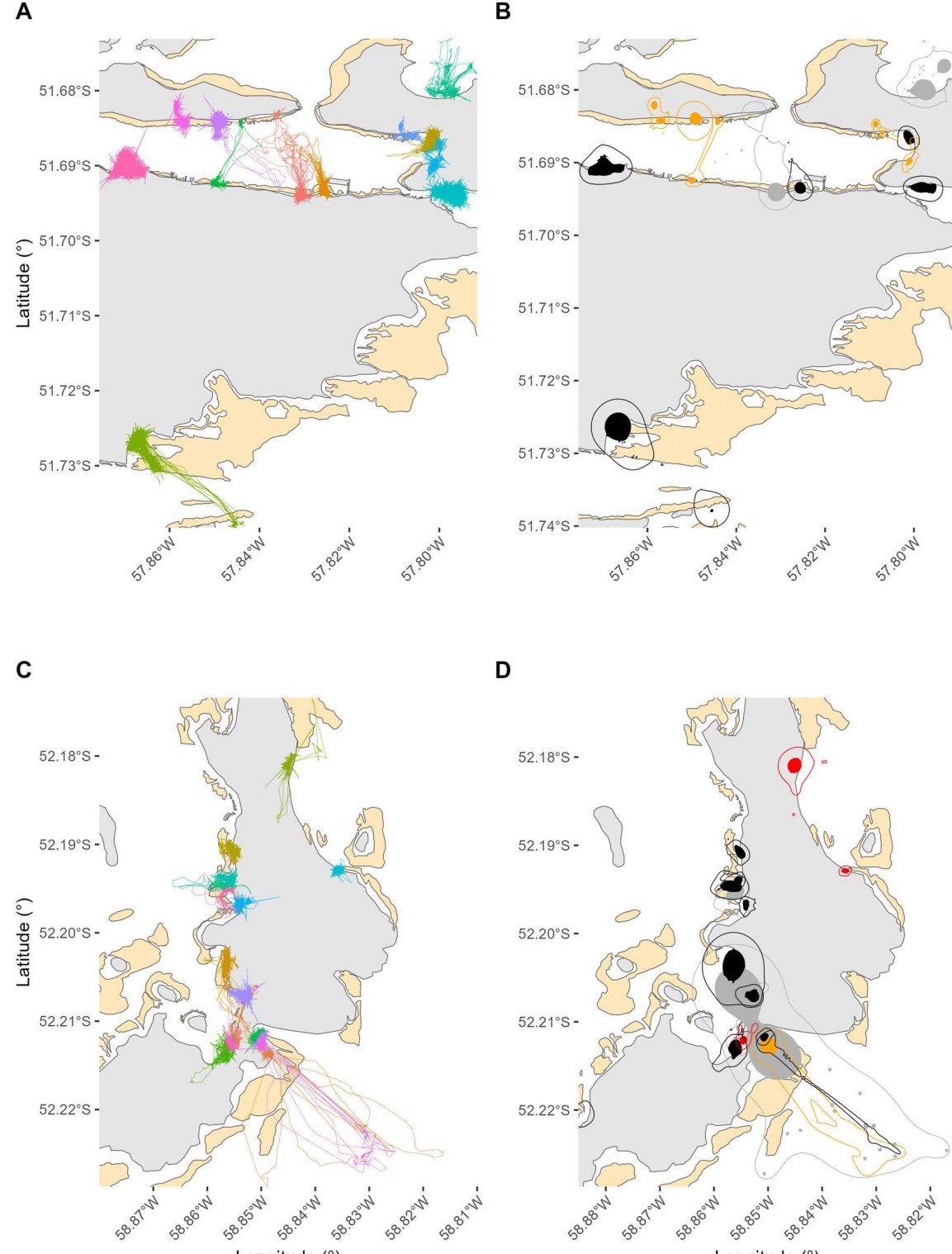

**Fig 2. Movement tracks of birds tagged on Stanley Harbour (A) and Bleaker Island (C), with their associated home and core ranges (respectively B and D).** Home and core ranges with sufficient data were coloured in black (males) and red (females) while nominal home and core ranges with insufficient data were coloured in light grey (males) and orange (females). Kelp beds were represented in beige.

[37]. Movement tracks were then interpolated at 2 min intervals using the *adehabitatLT* package [38] and daily distances travelled (km) were calculated along the trajectory of movement.

Home and core ranges were estimated by analysing the collected GPS data with the biased-random bridge kernel method [39,40]. Firstly, to identify resident individuals for whom we had sufficient data to calculate the home range, we produced variograms using the '*variogram*' function from the *ctmm* package. Individuals were considered resident when the variograms approached an asymptote [41]. Any individuals that did not reach or maintain an asymptote were not considered for home and core range analysis. We defined home range based on two levels of the Utilisation Distribution (UD) obtained via the biased-random bridge kernel method [39]. This method produces an occurrence distribution, rather than a true extrapolated home range, and represents a version of Brownian bridges [42] using the movement-based kernel density estimation. We estimated the wider home range ($UD_{95}$) and core range ($UD_{50}$), representing the area in which the individual can be found 95% and 50% of the tracked time. All metrics were extracted using '*getverticeshr*' function from the *adehabitatHR* package and the obtained contours mapped using the *ggplot2* package. To identify factors influencing the size of the core and wider home ranges, we ran, for each level, two linear regressions with either sex *or* breeding status, as well as proportion and absolute area of kelp, and study site, as fixed effects, and weighted by tracking duration, to account for differences among individuals in the length of time they were tracked.

To assess which factors influence daily distance travelled, we ran a Generalized Linear Mixed Model (GLMM; gaussian distribution, *glmmTMB* package), with fixed effects of breeding status, location, distance to kelp and to roads, and a random effect of individual.

Time spent on land was computed as the number of GPS locations on land. To identify factors influencing time spent on land, we used a GLMM (betabinomial family weighted by the total points at land and at sea), with breeding status, and study site as fixed effects and individual as a random effect.

From the GPS locations of each individual, three ecologically relevant statuses (resting, foraging and travelling) were determined using Hidden Markov Models. These models rely on an observable set of data (here the tracking data for each individual), to infer a non-observable state dependent on the distance and angle between subsequent points (step length, and turning angle) [43]. Using the *moveHMM* package [44], we defined the form of each state, which were then used to estimate the proportion (%) of time spent in each state. Resting is depicted as having the lowest step length and low turning angles. Commuting is best described by rapid movements, that is to say long step length and low turning angles. Whereas sharp turning angles and lower step lengths indicating more tortuous movements, indicative of exploration, represent foraging.

To identify factors influencing variation specifically in the proportion of time spent foraging per hour of the day, we ran a GLMM (beta-binomial distribution, weighted by total foraging points), with breeding status, location, distance to roads, settlements and kelp, as well as absolute kelp bed area as fixed effects, and individual as a random effect. The '*nearest*' function in the *terra* package in R was used to compute all distances. Distance to infrastructure (a proxy for human disturbance) was calculated as the distance between the centre of a given home range or each GPS location and the nearest road and settlement separately. Similarly, distance to the kelp was computed as distance from the kelp polygons (source: FK-SAERI-284) to coast, for all models using this variable, except for the proportion of time spent foraging. Instead, distance was computed between each GPS location and the nearest kelp forest. The proportion of kelp was calculated as the size of the intersection between the polygons of kelp beds and either the core or home range. Kelp area was calculated as the absolute value of area of kelp present in either the core or home range. Maps were made using the coastline shapefile provided by SAERI (FK-UKHO-414) and the *rnaturalearth* package [45].

For all models, track duration was added as a weight when relevant, and fit was assessed by simulating residuals in the *DHARMa* package. For each model, we have also included a null model with no additional terms to provide a baseline output. Terms were dropped sequentially, and models were ranked by AIC ('*dredge*' function, *MuMin* package), with a

minimum difference of ΔAIC = 4 [46]. If numerous candidate models were within ΔAIC = 4, we judged them to have equal support and performed model averaging ('*model.avg*' function, *MuMin* package). For each averaged model, the 95% confidence interval was calculated and effects that did not cross 0 were considered significant.

## Results

After assessment of the individual variograms (see Methods, and S1/ S2), 17 individuals were kept for home and core range analysis. As this resulted in too few incubating females and patrolling males, we only compared changes in home and core ranges between chick-rearing and non-breeding individuals. Overall, mean home range size was 19.36 ± 6.19 ha. Mean core range size was 3.44 ± 0.92 ha. Daily travelled distances were computed along each individual's trajectories. Falkland Steamer Ducks travelled on average 10.19 ± 0.50 km with a maximum of 13.52 ± 0.74 km and varied between breeding status (Table 1). In our preferred, lowest AIC model, all variables except status and proportion of kelp were retained for the home range analysis when status is considered (Table 2 and S1).

### Home and core ranges

Size of core and home ranges were hypothesised to be influenced by sex/breeding status, study site, absolute area of kelp, proportion of kelp and distances to roads, settlements and ponds. Mean core range size was not significantly influenced by sex (P = 0.595) nor by breeding status (P = 0.377). Kelp cover was significantly influencing the size of both home ($P_{sex}$ < 0.001; $P_{status}$ = 0.001) and core range ($P_{sex}$ = 0.001; $P_{status}$ < 0.001), unlike the proportion of kelp present (home range: $P_{sex}$ = 0.727, core range: $P_{sex}$ = 0. 117; $P_{status}$ = 0.377 –Fig 3). Study site was always significant, except when considering the core range between non-breeding and chick rearing individuals (Table 2 and Fig 3, Fig.4).

**Table 2. Model estimates for each dependent variable in Table 1. *=model averaged estimates. The values represent the 95% interval. Bold numbers show significance. Light grey indicates variables not kept for the averaged modelling and dark grey not included in the modelling.**

| Model | Intercept | Sex (male) | Status (IF) | Status (PM) | Status (CR) | Study site (Stanley) | Area kelp | Percent kelp | Distance to kelp (km) | Distance to roads (km) | Distance to settlement (km) | Distance to ponds (km) |
|---|---|---|---|---|---|---|---|---|---|---|---|---|
| Distance travelled (km.d⁻¹)* | −0.84– 4.21 | | **−6.51 – −0.09** | **−7.65 – −1.46** | −4.63– 0.83 | −0.05– 5.75 | | | **0.0005– 0.0141** | **0.0004 − 0.0007** | **0.0005 − 0.0029** | **1.42.10⁻⁶ 3.30.10⁻⁶** |
| Home Range (95%) – sex* | **1.20– 2.78** | −1.21– 1.76 | | | | **0.16–1.58** | **0.04–0.14** | 0.06–0.04 | | | | |
| Home range (95%) – status | **1.08 – 2.15** | | | | | **0.37–1.86** | **0.04–0.13** | | | | | |
| Core Range (50%) – sex* | −0.36– 0.83 | −0.92– 1.61 | | | | **0.41–1.58** | **0.30–1.16** | −0.05–0.01 | | | | |
| Core range (50%) – status * | −0.32– 0.67 | | | | [NB] −0.36– 0.94 | −0.06– 1.36 | **0.58–1.44** | −0.04–0.01 | | | | |
| Proportion time on land* | −1.44– 0.09 | | **0.65 – 3.23** | −1.98 – 0.71 | −1.06– 1.21 | −1.65– 0.28 | | | | | | |
| Proportion time foraging* | **1.27– 10.84** | −0.59– 0.56 | **0.04– 1.09** | −0.11– 0.96 | **0.11– 1.09** | **−9.42 – −1.26** | **50%: −1.30 – −0.33/ 95%: 0.05–0.22** | 50%: −0.03– 0.04/ 95%: −0.05–0.00 | −0.73– 0.47 | **−0.0004 – −0.0001** | **−0.61 – −0.21** | |

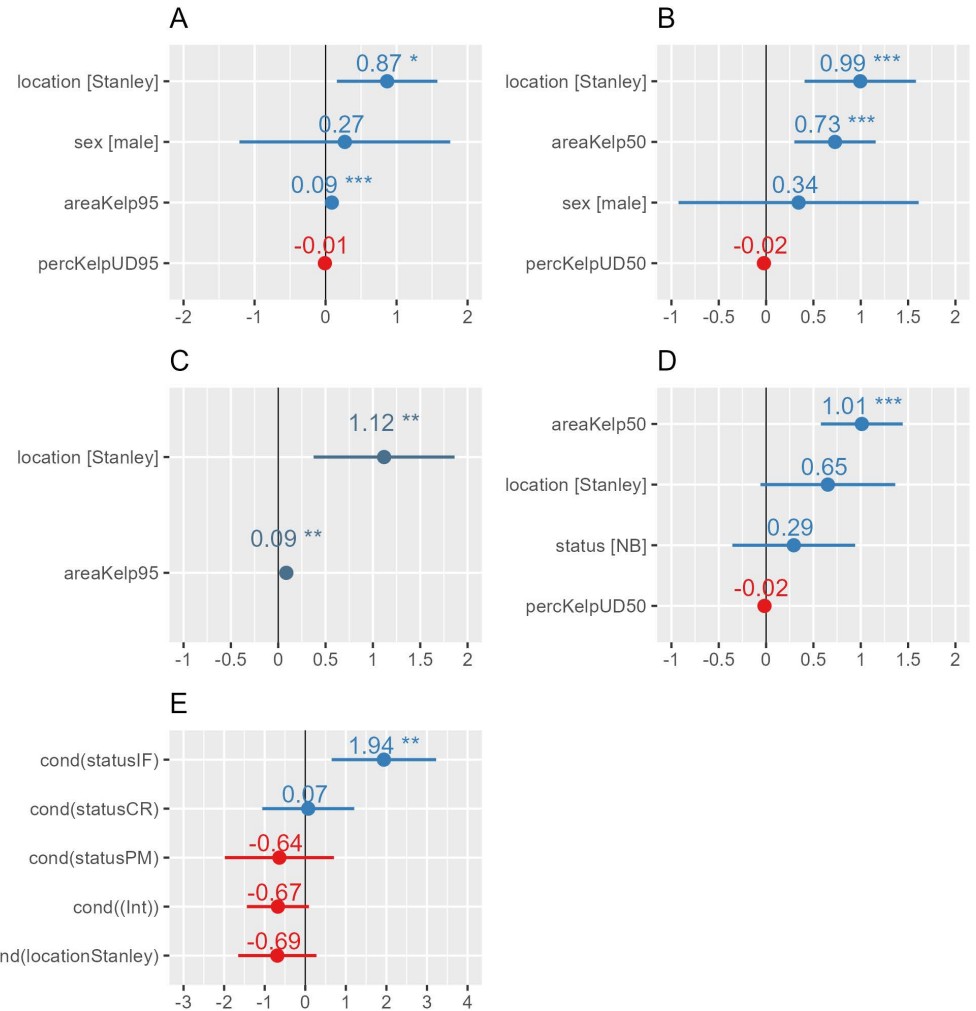

**Fig 3. Means and 95% confidence intervals of each predictor within each of the models; A: Home range (sex); B: Core range (sex); C: Home range (breeding status); D: Core range (breeding status); E: Time spent on land.**

## Daily travelled distances

Mean distance travelled was hypothesised to be influence by breeding status, study site and distance to kelp, roads, settlements and ponds. All considered variables were retained (Table 2 and Fig 3,Fig.4). Travelled distance significantly differed between breeding status, with incubating females (8.52±0.96 km.d$^{-1}$) and patrolling males (8.04±0.04 km.d$^{-1}$) travelling significantly less than non-breeding individuals (11.41±0.75 km.d$^{-1}$; P=0.004 and P=0.044 respectively). Travelled distances also increased significantly with greater distances to kelp beds (P=0.034) and to ponds (P<0.001). Likewise, the greater the distance to roads and settlements, the more individuals travelled (P<0.001 and P=0.004 respectively).

## Time spent on land

Proportion of time spent on land was hypothesised to be influenced by breeding status and study site. Similarly, all variables included in the analysis of time spent on land were retained (Table 2 and Fig 3,4). Time spent on land differed significantly

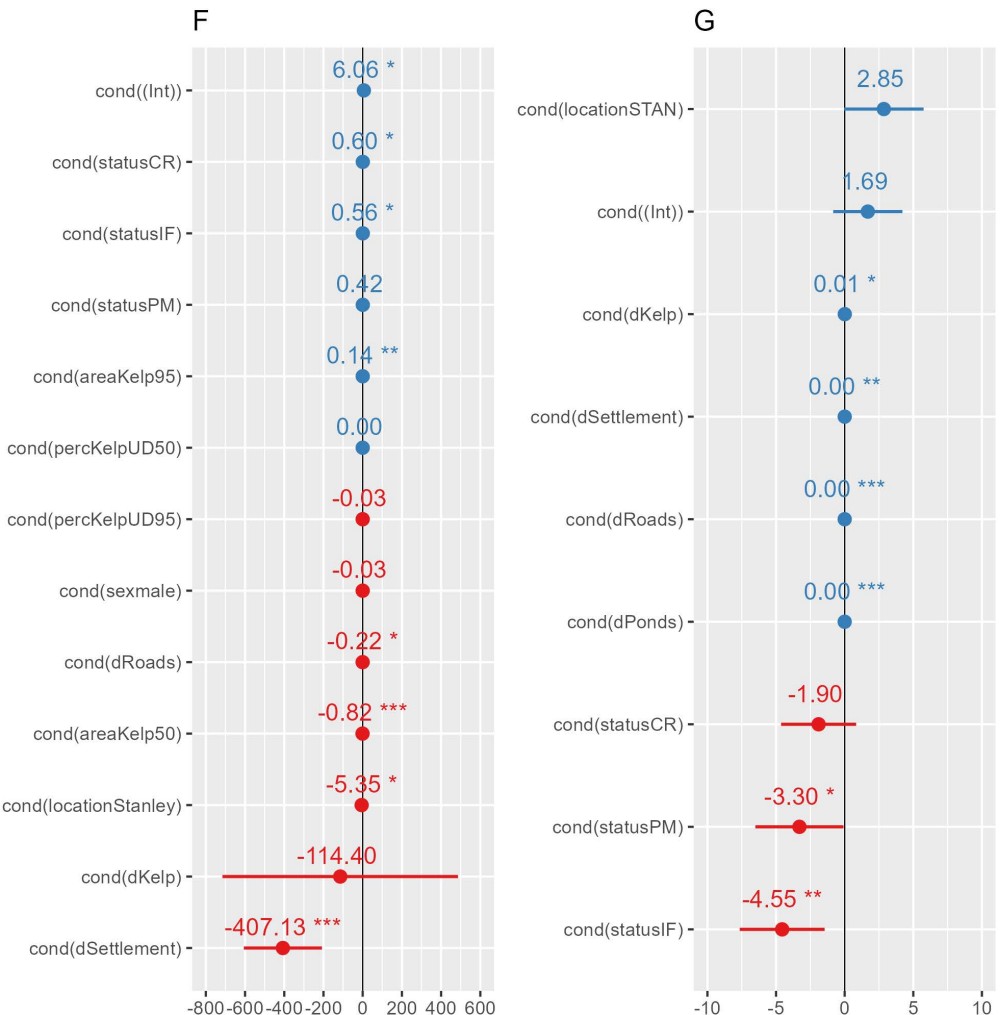

**Fig 4. Means and 95% confidence intervals of each predictor within each of the models F: Time spent foraging, G: Mean daily distance travelled.**

between incubating females (70.00±2.43% of recorded GPS points – P=0.003) and the rest of the breeding categories (NB: 32.50±1.73%; CR: 40.09±2.70% and PM: 21.83±2.73% of recorded GPS points– S 3). Incubating females exhibited two minimums in time spent on land: one between 5 and 7 AM and one between 6 and 7 PM (Fig 5). Chick-rearing individuals exhibited two declines in time spent on land around midnight, followed by a prolonged period between 1 AM to 11 AM during which they spent up to 63% of their time on land. A second minimum occurs between 12 AM and 4 PM. Non-breeding individuals showed a limited period on land, regardless of the time of the day. Patrolling males showed minimal land use around 9 AM, which increased to 42.5% at noon, followed by a gradual decrease until 8 PM. This was followed by a drop around midnight and a second peak at 4 AM, reaching a maximum of 45% before declining again. (Fig 5).

## Time spent foraging

Time spent foraging was hypothesised to be influenced by breeding success, study site, absolute kelp cover and proportion of kelp cover in both the core and home ranges and distances to roads, settlements, kelp and ponds. Distance

to ponds was the only variable not retained for the model averaging for time spent foraging. Individuals closer to roads (P=0.010) and settlements (P<0.001) spent significantly more time foraging. On the other hand, when considering the entire study sites, individual living in the area of Stanley Harbour spent significantly less time foraging (37.8±0.98%) than those living on Bleaker Island (54.0±1.25% - P=0.010). The wider the absolute kelp area in the core range, the less time was spent foraging while the reverse was predicted for kelp cover in the home range. On the contrary, neither the distance to kelp beds (P=0.667) nor the proportion of kelp beds influenced signifanlty at neither core (P=0.823) nor home (P=0.073) range level.

Collectively, individuals spent 21±1.8% of their time travelling, 48±2.8% foraging and 31±2.6% resting (Fig 6). Incubating females (P=0.036) and chick-rearing individuals (P=0.015) spent significantly more time foraging than patrolling males (P=0.123) when compared to non-breeding individuals. Sex did not have an effect, however (P=0.96). Incubating females spent 45.64±1.72% of their day foraging, chick-rearing individuals 55.93±1.72%, patrolling males 43.72±2.39% and non-breeding individuals 44.42±1.40%.

## Discussion

This study represents the first investigation into the movement ecology of Falkland Steamer Ducks, and more broadly of the *Tachyeres* genus, using high-resolution GPS tracking. Our results demonstrate that breeding status, study site, distance to ponds and kelp characteristics (*i.e.,* kelp cover and distance to kelp beds) all significantly influence different facets of Falkland Steamer Duck movement ecology. In addition, human disturbance also played a significant role. Our findings establish a baseline for understanding the spatial ecology of Falkland Steamer Ducks and highlight the species' potential role as a sentinel of environmental change.

### Home and core ranges

Home range sizes were significantly larger at Stanley harbour than Bleaker Island. Those differences in size might reflect differences in habitat quality. All home and core range models highlight the importance of absolute kelp cover and not of the percentage of kelp cover in either the home or core range. This suggests that the size of both the home and core

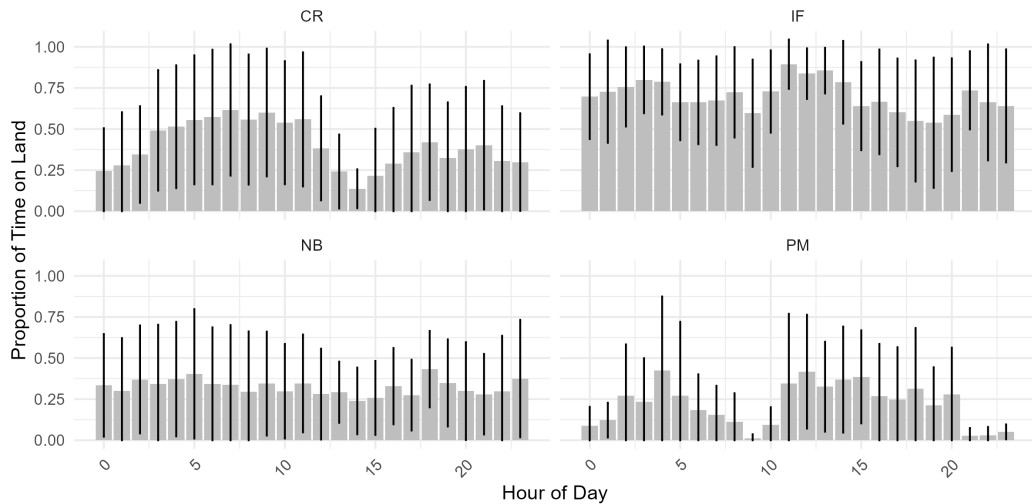

**Fig 5. Proportion of time spent on land per breeding status: Chick-rearing individuals (CR), Incubating females (IF), Non-breeding individuals (NB) and Patrolling Males (PM).** Proportions were calculated as the number of GPS locations on land divided by the total number of GPS locations per individual. Black lines indicate error bars.

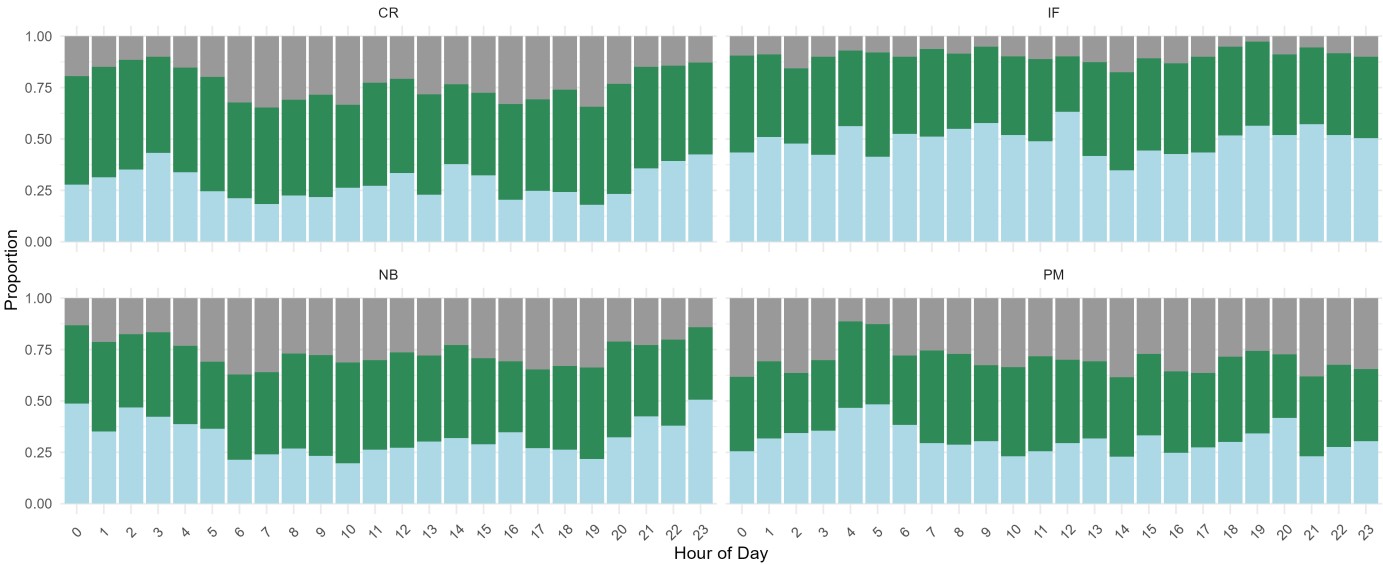

**Fig 6. Proportion of time spent resting (light blue), foraging (green) and travelling (light grey) per breeding status: Chick Rearing individuals (CR), Incubating Females (IF), Non-Breeding individuals (NB) and Patrolling Males (PM).**

range may vary to maintain access to a sufficient level of kelp beds. A larger territory may reflect the breeding pair's need to secure suitable kelp beds capable of providing adequate resources. Alternatively, differences in breeding pair density may play a role in territory size. For example, northern Bleaker Island supported a higher breeding pair density (9.8 pairs.km$^{-1}$) compared to Stanley (7.7 pairs.km$^{-1}$), which could reflect individuals occupying smaller, resource-rich territories (*Kristiansen et al in prep.*).

The proportion of kelp within the territory did not appear to influence either home or core range size, as opposed to absolute area of kelp. Several factors may explain this. First, territory size may vary to ensure access to a minimum threshold of absolute kelp cover. Once reached, the proportion of kelp becomes less relevant, as long as the kelp beds provide sufficient resources [47]. Second, Falkland Steamer Ducks may also rely on additional foraging resources beyond kelp beds. Individuals were frequently observed dabbling close to the shoreline [27], suggesting coastal nearshore habitat also provides access to prey resources. Further research into diet composition might shed light on the relative contribution of kelp-associated versus coastal food sources.

## Daily travelled distances

Incubating females and patrolling males travelled less than non-breeders. Within the *Tachyeres* genus, only females undertake incubation [15,36], with the male assuming sole patrolling duties during this period [20]. During incubation, females rely on an easy access to high-quality resources to minimize time away from the nest, and mitigate declines in body condition associated with maintaining optimal nest temperature [48,49]. Meanwhile, patrolling males have been described as surveying the sea territory near the nest [50]. Hence, the movements of incubating pairs are restricted, when compared to non-breeders.

The finding that duckling-rearing individuals had similar mean and maximum distance travelled as non-breeders likely reflects parents not needing to be central place foragers as ducklings can self-feed in the presence of adults throughout their territory [18,51]. It may also result from the parents holding a territory, which requires defending

regardless of the presence or not of ducklings. Anecdotally, a female with ducklings was seen joining her partner in a territorial fight, despite the potential negative effect of abandoning the brood (*e.g.,* injury due to the fight, opportunities for flying predators).

Individuals travelled more when the distance to kelp and to ponds increased, potentially reflecting the need for territories to include feeding and resting areas but also access to freshwater. In diving sea ducks, drinking freshwater was thought to reduce salt stress and suggested salt stress as possible structuring factor when considering habitat quality [28]. In our study, tracked individuals walked inland solely to drink. Additionally, individuals travelled more when distance to roads and settlements increased. One hypothesis could be that territories away from human structures might present higher-quality habitats, where more individuals decide to establish their territories. Where there is a higher density of individuals, more time might be dedicated to territory defence, and neighbouring birds may have to travel further to access resources and avoid conflict. This could also help to explain the high individual variation, and high residual variation (i.e., unexplained differences), in daily distance travelled. Unfortunately, true density of neighbours was not identified for each tracked individual. In the future, these data should be collected, and the use of accelerometery data could also identify territorial displays, such as steaming and physical contact with intruding neighbours.

## Time spent on land

Incubating females spent the most time on land, with minimums close to sunrise and sunset. More time spent on water at these times could translate a trade-off between the need to forage and the risk of nest predation by diurnal predation. A similar pattern was found for the Chubut Steamer Duck [50] and was hypothesised to represent a strategy to avoid nest predation from visual predators such as the Kelp Gull (*Larus dominicanus*), and the Striated caracara (*Phalcoboenus australis*).

All tagged parents had, at the time of the monitoring, young ducklings. Due to their downy plumage and lack of efficient insulation, ducklings present a limited waterproofness making them more prone to lose heat and unable to stay in water for extended periods of time [52]. As the ducklings grow older, their plumage retains less water, meaning they can retain their body heat, allowing them more time on water [53]. Hence, parents with young ducklings are more temporally constrained to shore than those with older ducklings, and non-breeding individuals [50] (Fig 5). Indeed, non-breeding individuals appear to be spending on average more than 60% of their time at sea, which could originate from the necessity to prevent intrusion from juveniles [15] or neighbours [26] through patrolling. This translates in the mean distance travelled per day. Both non-breeding and chick-rearing pairs are at a stage where they can defend their access to kelp. Non-breeding pairs only differ in their time spent on land since they are not constrained by a brood.

Patrolling males also spent most of their time at sea. During incubation, the male is known to patrol mainly in waters in front of the nest and swim closely to its female while feeding [20]. This resembles territorial behaviour, where the male aggressively defends the area around the female [21]. This behaviour may persist during chick-rearing and non-breeding stages, though the female is also known to defend their home range [26].

## Time spent foraging

Overall, individuals living closer to settlements and to roads spent more time foraging. Such individuals are more likely to be disturbed by human activity, therefore decreasing time spent resting to compensate for foraging time loss, but also spending an increased amount of time being vigilant and moving away from humans or dogs, which could be confused with foraging behaviour in the GPS data. This explanation is supported by fieldwork observations in which Falkland Steamer Ducks, when disturbed by human activity while resting on shore, were seen swimming away and resuming foraging. The disturbance effect of recreational activity has been measured in seven different wintering duck species in the Back Bay National Wildlife Refuge, Virgina Beach, USA [33]. Unfortunately, quantifying the effects of human disturbance remains challenging, especially in a data-limited environment such as the Falkland Islands [32].

Individuals on Bleaker Island were found to spend more time foraging than those around Stanley Harbour. This was unexpected, as four individuals were living in the city itself and all but one incubating female were either nesting close to or required to cross a road to reach feeding grounds. We therefore expected a higher proportion of foraging activity among individuals in Stanley Harbour compared to those on Bleaker Island. Bleaker Island was characterised by one settlement and no concrete road. The nature of the coastline varied but human disturbance was low. On the other hand, Stanley Harbour presented a variety of human infrastructures, from occasionally visited jetties, to the city of Stanley. Disturbance therefore varied greatly depending on the specific location of each tagged individual. This highlights the slightly unreliable result from using a binary location variable as an effect in the model, as opposed to, finer-scale proxies such as distance to roads or settlements, which may offer more reliable indicators of foraging activity.

Incubating females and chick rearing individuals spent more time foraging. These individuals are likely to require more energy as they are actively involved in reproduction, and lose energy when incubating and brooding [49,50]. In our study, all females were less than 100 m away from a kelp bed (Fig 2), and this short travel distance means that they could potentially spend more time foraging before returning to their nest. On the other hand, chick-rearing parents displayed a higher frequency of foraging (Fig 6). This further supports the idea of limiting heat loss for the young ducklings while ensuring they gain the energy required for an optimal growth [53]. Time spent foraging was higher where there was more kelp cover in the wider home range but lower where there was more kelp cover within the core range.

## Conclusion

This study highlighted variations in the movement ecology of the endemic Falkland Steamer Duck based on breeding status, sex, and in relation to environmental factors. Distance to human structures, used as a proxy for human disturbance, were found to affect distance travelled and time spent foraging by Falkland Steamer Ducks. Kelp beds also constrained Falkland Steamer Ducks, from the size of their home and core range to their daily activity. Daily travelled distance also increased if ducks were required to travel inland for the purposes of accessing freshwater. Breeding status constrained time spent foraging and time spent on land.

Kelp, and more specifically the dominant *Macrocystis porifera*¸ are engineering species which are found in dynamic coastal regions that provide nursery areas for numerous marine species such as Patagonian squid (*Doryteuthis gahi*), rock cod (*Patagonothen* spp) and the Southern blue whiting (*Micromesisitius australis*), and therefore provide a link with open-sea trophic webs [32,54,55]. Based on our observations and literature [18], we were expecting Falkland Steamer Duck ecology to be influenced by kelp forest as their main food source. Given that the Falkland Steamer Duck is an endemic species distributed along the entire coastline of the Falkland Islands archipelago, it holds considerable potential as an indicator species for environmental changes both terrestrial (*e.g.*, coastal erosion, shifts in *Poa flabellata* [tussac] density, drying up of ponds) and marine (*e.g.*, degradation of kelp beds). Obtaining reliable ecological information is therefore critical for accurately assessing the conservation status of the species and for developing effective management strategies to ensure its persistence—particularly in the face of climate change and its associated impacts on kelp ecosystems. The future of regional oceanographic conditions remains uncertain, with the South Atlantic Ocean experiencing warming trends while the Falkland Current appears to be cooling [32]. Such changes could have cascading ecological consequences, particularly for kelp forests, which, while historically stable [54,55], may be vulnerable to abrupt shifts. Understanding thoroughly what induces the species distribution and habitat selection throughout the entire archipelago coastline is a key step toward predicting the cascading impact of climate change on the Falkland Steamer Duck through any changes of key environmental features of the marine/terrestrial coastline interface. Continued monitoring will be essential to detect emerging ecological shifts and to inform adaptive conservation strategies in this rapidly changing region.

## Supporting information

**S1 Fig. Variograms for individuals with insufficient data to estimate home ranges.** Following Calabrese et al (2016) recommendation, because the curve does not reach an asymptote, the showed individuals were excluded.
(DOCX)

**S2 Fig. Variograms for individuals with sufficient data to estimate home ranges.** Following Calabrese et al (2016) recommendation, because the curve does reach an asymptote, the showed individuals were kept.
(DOCX)

**S1 Table AIC and delta AIC for models explaining (i) travelled distance, (ii) home range, (iii) core range, (iv) proportion of time spent on land, (v) proportion time foraging.** (1/id) notation represents individual as a random effect. **Bold** models are single models with the best score, or multiple models used for model averaging (those within delta AIC = 4). Null models (i.e., no covariates) presented in italics.
(DOCX)

**S3 Fig. Activity budget of the Falkland Steamer Duck.** Colours represent the different behaviours (grey: travelling; green: foraging; blue: resting) for each breeding status (CR: chick-rearing; IF: incubating female; NB: non-breeding; PM: patrolling male).
(DOCX)

## Acknowledgements

We sincerely thank the South Atlantic Environment Research Institute for the logistic support and access to field. We also extend our gratitude to Nick Rendell and his family for their warm hospitality on Bleaker Island. We are appreciative of all the feedbacks from Dr María Laura Agüero and reviewer 1 which strengthened the structure of this publication.

## Author contributions

**Conceptualization:** Alix M.I. Kristiansen, Alastair M.M Baylis, John P.Y. Arnould.

**Formal analysis:** Alix M.I. Kristiansen.

**Funding acquisition:** Alix M.I. Kristiansen, John P.Y. Arnould.

**Investigation:** Alix M.I. Kristiansen, Alastair M.M Baylis, Sébastien A.P. Dupray, Heather Q. Mathews, John P.Y. Arnould.

**Methodology:** Alix M.I. Kristiansen, Sébastien A.P. Dupray, Lucy J. Mitchell, John P.Y. Arnould.

**Resources:** Alastair M.M Baylis.

**Supervision:** Luc Lens, John P.Y. Arnould.

**Writing – original draft:** Alix M.I. Kristiansen.

**Writing – review & editing:** Alix M.I. Kristiansen, Alastair M.M Baylis, Sébastien A.P. Dupray, Luc Lens, Lucy J. Mitchell, Heather Q. Mathews, John P.Y. Arnould.

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
