## [Decision Letter · Decision Letter 0]

26 Jun 2025

PONE-D-25-23682Habitat use and activity budget in the Falkland Steamer Duck (Tachyeres brachypterus)

PLOS ONE

Dear Ms. Kristiansen,

Thank you for submitting your manuscript to PLOS ONE. I have now received reviews of the paper from two experts who are knowledgeable about sea ducks, including this species, and they have made a number of constructive suggestions about how to improve the paper and communicate your Ph.D. study results. So, at this point, the paper is not suitable for acceptance, but I think the two reviewers have given you a lot to work with and I would encourage re-submission of a revised version once you have considered their recommendations.  

We look forward to receiving your revised manuscript.

Kind regards,

Lee W Cooper, Ph.D.

Section Editor

PLOS ONE

Journal Requirements:

LL and LM’s work was partially supported by the Methusalem project (01M00221). Fieldwork was funded by the Shackleton Scholarship Fund (SSF22-019-ADC-Kristiansen and SSF23-021-ADC-Kristiansen) and the Falkland Environmental Study Budget (ESB122022).

LL and LM’s work was partially supported by the Methusalem project (01M00221). Fieldwork was funded by the Shackleton Scholarship Fund (SSF22-019-ADC-Kristiansen and SSF23-021-ADC-Kristiansen) and the Falkland Environmental Study Budget (ESB122022).

LL and LM’s work was partially supported by the Methusalem project (01M00221). Fieldwork

was funded by the Shackleton Scholarship Fund (SSF22-019-ADC-Kristiansen and SSF23-

021-ADC-Kristiansen) and the Falkland Environmental Study Budget (ESB122022). We

sincerely thank the South Atlantic Environment Research Institute for the logistic support and

access to field. We also extend our gratitude to Nick Rendell and his family for their warm

hospitality on Bleaker Island.

 LL and LM’s work was partially supported by the Methusalem project (01M00221). Fieldwork was funded by the Shackleton Scholarship Fund (SSF22-019-ADC-Kristiansen and SSF23-021-ADC-Kristiansen) and the Falkland Environmental Study Budget (ESB122022).

No.

7. Please amend the manuscript submission data (via Edit Submission) to include author Sébastien Dupray.

8. Please amend your authorship list in your manuscript file to include author Sébastien A.P. Dupray

9. Please ensure that you refer to Figure 3 in your text as, if accepted, production will need this reference to link the reader to the figure.

10. Please include a caption for figure 5.

11. We note that Figures 1 and 2 in your submission contain map images which may be copyrighted. All PLOS content is published under the Creative Commons Attribution License (CC BY 4.0), which means that the manuscript, images, and Supporting Information files will be freely available online, and any third party is permitted to access, download, copy, distribute, and use these materials in any way, even commercially, with proper attribution. For these reasons, we cannot publish previously copyrighted maps or satellite images created using proprietary data, such as Google software (Google Maps, Street View, and Earth). For more information, see our copyright guidelines: http://journals.plos.org/plosone/s/licenses-and-copyright.

a. You may seek permission from the original copyright holder of Figures 1 and 2 to publish the content specifically under the CC BY 4.0 license.  

12. Please include a copy of Table 3 which you refer to in your text on page 7.

13. Please include captions for your Supporting Information files at the end of your manuscript, and update any in-text citations to match accordingly. Please see our Supporting Information guidelines for more information: http://journals.plos.org/plosone/s/supporting-information.

Reviewers' comments:

Reviewer's Responses to Questions

**Comments to the Author**

1. Is the manuscript technically sound, and do the data support the conclusions?

Reviewer #1: No

Reviewer #2: Yes

2. Has the statistical analysis been performed appropriately and rigorously? 

Reviewer #1: Yes

Reviewer #2: Yes

3. Have the authors made all data underlying the findings in their manuscript fully available?

Reviewer #1: Yes

Reviewer #2: Yes

4. Is the manuscript presented in an intelligible fashion and written in standard English?

Reviewer #1: Yes

Reviewer #2: Yes

5. Review Comments to the Author

Reviewer #1: This paper provides new descriptive information on an interesting and little-studied species. The paper would be improved if a better context were provided to motivate collection of the descriptive data. The study is justified as collection of baseline information, but the usefulness of data on home range sizes and activity budgets depends on the types of change anticipated and how the data could be applied to problems that might arise. Distance to kelp and distance to human structures were logical candidates for possible change. However, finding that the birds traveled farther when distance to kelp was greater does not imply constraint when the areal extent of kelp in the home range had no effect, and distance to human structures does not seem a problem because the birds fed more the closer they were to structures. The overall conclusion that climate-driven changes in kelp availability are expected to negatively affect the species is not derived from or justified by analyses presented in this paper, which showed that kelp availability had little effect on movements or home range size.

The manuscript could also be more carefully prepared, as noted in the comments below. I have made a number of suggestions that might help improve the paper.

1. L 10. Replace “conversation” with “conservation”

2. L 19-20. Insert “birds of different” before “breeding status”

3. L 21-23. This conclusion is not logical or compelling as stated. The sentence first states that kelp beds occur “all over the archipelago” and are thus perhaps far from limiting in availability, but then warns that any changes in density of kelp beds are “expected to negatively impact the species”. To draw such a conclusion, some quantification of the area of this habitat available vs. the area of this habitat occupied by the current steamer duck population is needed.

Moreover, what do you mean by kelp “density”? Fraction of total area occupied by kelp habitats? Biomass of kelp per unit area of habitat? This qualitative conjecture could have been stated anecdotally without conducting the study, so please back it up with specific analyses.

4. In general, the Abstract should contain more specific results. What were the distances traveled and home and core range sizes among birds of different pair status (or at least the ranges observed). What were percentages of time spent in different behaviors? What specific variables led you to your final statement that changes in the density of kelp beds would negatively the species? I suggest substantially reducing the first half of the Abstract which is all introductory, and then providing some specific, quantitative findings. The Abstract is important for people who will not read the paper, so you need to provide more substance.

5. L 32-33. The passage should read “… invasive species, climate change, and inaccessibility often …”

6. L 34-40. This paragraph adds little that ecologists reading the paper do not already know. I suggest deleting the paragraph.

7. L 41-44. The first sentence of this paragraph can be deleted with no loss of information important to this paper. Eliminate summaries of general knowledge and get to the substance of specific questions you will address in this study.

8. L 47. Explain what you mean by “passive defence”

9. L 49-50. To be more concise, I suggest ‘Territories of marine waterfowl encompass both marine and terrestrial habitats.”

10. L 53. I strongly suggest that you avoid acronyms unless the terms are really ponderous. Busy scientists often do not have time to read a paper in one sitting within the same few days, and may skip around in a paper to find particular information. Consequently, it is quite annoying have to search back through a paper to find what a particular acronym stands for.

11. L 57. Replace “apparent” by “apparently”. The pairbonds are not apparent, they are apparently long-term.

12. L 60. Replace “animal” by “bird and mammal” if that’s what you mean. I suspect there are endemic insects, arachnids, or other invertebrates that you have not considered in this general statement about animals.

13. L 63. I suggest deleting “interfacing” as unnecessary, and replacing “comprise” by “include”

14. L 69. Do you mean “these factors”, referring to sex, breeding status, and location mentioned in the preceding sentence? Why would you expect these factors to result in differences in these activities? You have not commented on potential reasons in preceding text, but such expectations could provide you with more specific predictions (and related conclusions) based on ecological processes of interest to a general readership.

15. L 83-84. How far inland do the ducks nest?

16. L 117. Please state clearly whether “daily distance traveled” was along the trajectory of movement rather than the maximum linear distance moved between the beginning and end of the day.

17. L 118-119. For the many readers who will not know, please explain briefly what the “random bridge kernel method” does.

18. L 120. What does “UD” stand for? An intuitive variable name is desirable.

19. L 123. Again, please explain in one or two sentences how “Hidden Markov Models” work.

20. L 124. Delete “in each of three ecologically relevant states (i.e.”. Based on GPS readings every 2 min, please explain how you discriminated resting vs. foraging in one place, and moving among nearby foraging patches vs. commuting.

21. L 148. Two days seems inadequate for defining an individual’s home range. Did you examine the data to see after how many days the home ranges mostly stabilized? Some standards for adequate sample size (in days) seem important to unbiased estimates of true home range size.

22. Figure 2. Panels A, B, D, and E are nice depictions of the ducks’ movements. However, the green areas that supposedly show foraging areas in panels C and F are effectively impossible to see, even when scrutinized with a magnifying glass. A reader with red-green color blindness would have no chance of discriminating what is shown in this figure. I suggest that panels C and F be moved into a separate figure, enlarged, and alternative colors selects (perhaps yellow for foraging).

What is represented by the areas in beige?

23. Table 1. Please state clearly in the caption whether the distances traveled were al What ong the trajectory of movement or were the linear distance between points at the start and end of the day. Please also specify “km/day” (not just km) as the units in the table heading. How did you standardize the core and home range sizes among individuals that were tracked for a little as 2 days to as much as 43 days (see L 148). Ranges of the birds are likely to differ substantially between such short and long tracking periods. Without better explanation of how you standardized and calculated these values, it’s not clear that results for different groups can be directly compared.

24. Tables 1 and 2 are both labeled as Table 1. “Dependent” is misspelled in what should be Table 2.

25. Table 2. Here and elsewhere in the paper, I suggest replacing “location” with “study site” to clarify what you mean (if in fact that’s what you mean). In this paper, you have locations every 2 min via the GPS, but two different study sites.

I also suggest “dist to kelp” and “dist to structure”, which are not much longer and clarify what the variable represents.

26. Table 2. The second and third variables for Distance per day are exactly the same, yet you assign them different values of AIC and ΔAIC.

27. Table 2. “hour.FI” (under Proportion of time on land) has not been defined as a variable, and I cannot guess what it might be.

28. What should be Table 3 is labeled as Table 2. Please designate the units for Distance traveled as km/day (not just km), and spell out the variable “Int”.

29. P 17. The authors stopped numbering lines at the beginning of P 17, and in fact do not include page numbers for any pages in the manuscript. These omissions make it harder to reference particular lines of text.

30. P 17, par 2, L 2. Please be specific in use of the term “home range” and “territory”, as they have different meanings. A home range is simply an area occupied, whereas a territory is actively defended. If in this sentence you mean that the non-breeding pairs defended these areas, replace “held” by “defended”. If they were occupying these areas without defense, then replace “held” by “occupied”.

31. P 17, par 4, L 1. Figure 4 (cited at the end of this sentence) shows only effects of breeding status, so delete “time of day and” from L 1. Time of day is dealt with in the last sentence of the paragraph.

32. The captions for Figures 3 and 4 at the bottom of P 17 are correctly numbered, but the figures cited as 4 and 5 in the preceding paragraph should be cited as 3 and 4, given that the original Fig. 3 included with the paper is never cited or mentioned in the manuscript.

33. P 18, par 3. I suggest that you do not attribute importance to differences that you found not to be significant, without citing actual P-values or effect sizes.

34. P 19, L 3. Delete “breeding”, or else use “sex of breeding birds” if that’s what you mean.

35. P 20, par 2, L 6. Please briefly explain the concept of “dear enemy”, which will be unfamiliar to many readers.

36. P 20, par 2, L 9. Replace “mean in” with “means by”

37. P 21, par 3, L 6. Insert “by” before “kelp forest”

38. P 21, par 3, L 9. “patting”? I have never seen this term applied to birds, so please define or substitute a more widely recognized term.

39. P 21, last 2 lines. You have presented no evidence that kelp is limiting to these ducks, and in fact your data indicate that sizes of their home ranges or defended areas are unaffected by the local availability of kelp. You have presented no evidence that these ducks forage preferentially in kelp beds. Perhaps you can cite other studies that have data to show such relationships, but you have not mentioned them.

40. The Discussion section on Time spent foraging and the Conclusion contain a fair amount of speculation about factors not really addressed in this paper. I suggest sticking to arguments for which you present more relevant data.

Reviewer #2: General Comments

This manuscript, "Habitat use and activity budget in the Falkland Steamer Duck (Tachyeres brachypterus)," presents novel findings for this endemic sea duck species in the Falkland Islands, addressing significant information gaps.

I believe its publication is highly important, as it provides systematic and rigorous data that can be valuable for zoning and management plans in the area. However, the authors need to organize the information more effectively to enhance readability and understanding. They propose analyses that are not clearly specified in the aims, the results section lacks full organization, and at least two figures are illegible.

Given the emphasis on the importance of kelp beds for the species (which I agree with), this topic should be more thoroughly introduced. I recommend adding references related to the FSD's diet. Specifically, I suggest reviewing Livezey (1989), "Feeding morphology, foraging behavior and food of Steamer-ducks (Anatidae: Tachyeres)," Occasional Papers of the Museum of Natural History, University of Kansas.

Introduction

(Line 54-55) “…..found solely in South America and the FLK (25,26)”

Suggestion: I suggest changing this to "southern South America," as the Falkland Islands are included within the continental shelf.

(Line 62) You could mention that the IUCN categorized this species as "Least Concern" based on a lack of detailed information, highlighting the importance of the current study.

(Line 61) “Such information is crucial as their marine-terrestrial interfacing territories comprise, amongst other features, kelp (mainly Macrocystis pyrifera (32)) and inland vegetation assemblages (33) which may be impacted by the predicted climate change increase of 1.8 °C before the end of the century (34).”

Suggestion: You might introduce some preliminary information about the feeding methods and diet of FSDs (Livezey 1989) here. This would help establish a clearer link to your hypothesis regarding the stronger presence of kelp beds within the species' core home range.

Methods

(Line 75) As you have described the characteristics of the coastline in Bleaker Island, I believe you must do the same for Stanley Harbour. This is especially relevant in light of Livezey (1989), where the author states that "...SD on both fresh and salt water were observed more frequently along shores dominated by rock outcrops and stony beaches, and less frequently on sandy or muddy shorelines..."

(Line 92) Breeding status categories:

How did you determine males were partners of incubating females? Were males patrolling the shoreline in front of nests where females were incubating?

Similarly, how did you determine the "non-breeding" category? Were these individuals grouped far from breeding pairs (a characteristic behavior of juvenile steamer ducks)?

How did you determine the sex of non-breeding individuals, especially given that non-breeding juveniles often have confusing plumage?

Suggestion: Perhaps a more appropriate categorization would be "Adults" (including incubating females and patrolling males) and "Juveniles."

(Line 101) How long did the GPS data logger track individuals (e.g., 1 month? 1 year?)? Instead of only specifying that data collection lasted until the logger was shed or the battery failed, you should specify the range/average time during which you tracked individuals.

(Line 104) I believe you must aggregate "...determine environmental factors that affect FSD behavior" within your aims section. Additionally, you should clearly specify the behavioral categories you had in mind for evaluating habitat use, activity budget, and the effects of environmental variables.

(Line 106-107) As mentioned previously, you need to introduce available information about the importance of kelp beds as feeding habitat for FSDs. You could cite Livezey (1989) here.

(Line 111-112) Perhaps a more appropriate variable would be the area of the kelp bed polygon, serving as a proxy for the amount of available food, given that kelp beds harbor a great diversity of steamer duck food items.

(Line 124) "Resting, foraging, commuting" are three categories of behavior, not ecologically relevant states.

(Line 126) Please specify this as an aim.

Several analyses you mentioned are not specified in the aims section. I think it is important to list them: "...factors influencing (1) FSD behavior, (2) daily distances traveled (km), (3) the size of the core and wider home ranges, (4) variation in the proportion of time spent foraging per hour of the day, and (5) time spent on land."

Do you consider time spent on land as resting behavior?

Pay close attention to the use of "home range" and "habitat use" terms.

"Home range refers to the spatial area that an animal or group of animals regularly uses to conduct all its normal vital activities. This includes foraging, reproduction, offspring care, resting, etc."

However, "habitat use refers to what kind of resources and features, present within its home range or within a broader area (e.g., the landscape), an animal or population utilizes to cover their needs. It's not just about where the animal moves, but what specific 'elements' of that place are used."

Your methods allowed you to determine home and core range, but not habitat use. To determine habitat use, you would need to sample the resources that FSDs use. For example, to determine breeding habitat use, you could focus on environmental features individuals use for nesting (microhabitat scale), such as vegetation, nest material, distance to the coastline, soil, etc.

Results

(Line 149-153) The "breeding status category" and the "amount of individuals of each sex within these categories" are not entirely clear until this paragraph. I think you should move this information to the methods section and clarify if the "non-breeding" category corresponds to juveniles or adults suspected of breeding season failure.

(Page 17) When you mention, “The hour of day was also significant (P < 0.01). Proportion of time spent on peaked above 0.50 at 4:00 and 11:00, then reached a minimum of 0.32 at 22:00,” are you referring to incubating females?

Does it have any biological sense to test the effect of the "incubating females" category on "time spent on land"? Perhaps I am misunderstanding the meaning of "time spent on land." Could it encompass resting, incubation, or patrolling? Does "on land" refer to the coastline, inland areas, or both?

Discussion

According to Table 2, breeding status (IF) and location were significant for travelled distance, and only sex (M) was significant for core range. The first paragraph of this section is unclear.

“In particular, we reveal that kelp distribution and distance to settlements influences FSD ecology, and more precisely its daily travelling distances and time spent foraging.”

“Home range sizes were larger at Stanley Harbour than Bleaker Island though not significantly. This may reflect birds needing to travel further to find suitable habitat.”

Why do you think this? Do you have any references to cite that support this explanation?

“Non-breeding pairs had the greatest variation between their home and core ranges, which could be linked to territorial behaviours such as patrolling and defence against intruders (31,48).”

What could be the possible explanation for the differences in home and core range between non-breeding individuals (assuming they are not juveniles) and "patrolling males"? That is, territorial behavior and aggressive defense are common during the breeding season. Therefore, this explanation does not sound like a suitable explanation for your results in this context.

“Core range was larger for males than females. This could originate from different behaviours. During incubation, the male is known to patrol mainly in waters in front of the nest and swim closely to its female while feeding (29). This resembles a form of territorial behaviour, where the male aggressively…”

Considering that your work was conducted during the breeding season, and adding that males patrol the marine coastal section in front of incubating females (Agüero and García Borboroglu 2013), "patrolling males" and "incubating females" could be considered as a "breeding pair" unit. In this case, home and core range could be obtained by the overlap between "patrolling males" and "their incubating females."

Home and core range are the places where all biologically important activities occur (feeding, breeding, resting, etc.). The fact that females spend more time on land and use this habitat while their male is patrolling the water territory has to do with the fact that only females incubate eggs. However, this does not mean that land used for nesting and water territory should be considered as different core ranges between sexes. I think the best way to test home range differences between sexes would be to track individuals outside the breeding season, or perhaps more appropriately, between age classes (pairs and juveniles), taking into account that steamer ducks are paired year-round and juveniles often group far from pairs.

(Page 21) “...For example, integrating the distance from kelp beds to the shoreline, and when possible, to nesting sites, may offer a more refined understanding of habitat use….”

Do you have the geolocations of kelp beds and nests of the incubating females you tracked? If so, you might consider adding this analysis to improve your manuscript, especially given that many of your potential explanations refer to the importance of kelp beds for FSDs.

(Page 22) “This study represents the first investigation into the movement ecology of FSD using high resolution GPS tracking. Our results demonstrate that breeding status, sex, geographic location, and kelp distribution all significantly influence different facets of its movement ecology. Our findings establish a baseline for understanding the spatial ecology of FSD and highlight the species’ potential role as a sentinel of environmental change.”

Suggestion: I agree; this paragraph would be a great sentence to head the Discussion section.

I could not understand the figures because they are in low resolution, so unreadable. Please improve them.

6. PLOS authors have the option to publish the peer review history of their article (what does this mean? ). If published, this will include your full peer review and any attached files.

**Do you want your identity to be public for this peer review?** For information about this choice, including consent withdrawal, please see our Privacy Policy .

Reviewer #1: No

Reviewer #2: **Yes: ** MARIA LAURA AGüERO

---

## [Author Response · Author response to Decision Letter 1]

4 Sep 2025

Reviewer #1:

This paper provides new descriptive information on an interesting and little-studied species. The paper would be improved if a better context were provided to motivate collection of the descriptive data. The study is justified as collection of baseline information, but the usefulness of data on home range sizes and activity budgets depends on the types of change anticipated and how the data could be applied to problems that might arise. Distance to kelp and distance to human structures were logical candidates for possible change. However, finding that the birds traveled farther when distance to kelp was greater does not imply constraint when the areal extent of kelp in the home range had no effect, and distance to human structures does not seem a problem because the birds fed more the closer they were to structures. The overall conclusion that climate-driven changes in kelp availability are expected to negatively affect the species is not derived from or justified by analyses presented in this paper, which showed that kelp availability had little effect on movements or home range size.

The manuscript could also be more carefully prepared, as noted in the comments below. I have made a number of suggestions that might help improve the paper.

We thank you for your feedback, your thorough read through and multiple suggestions. We certainly take into account the way that the information has been presented, and that we could have clarified elements better in order to present our arguments. We hope that the addition of the absolute kelp area helps to clarify the relationship between kelp beds and the species and that the manuscript now flows better and present stronger results.

1. L 10. Replace “conversation” with “conservation”

This has been changed. The sentence now reads as follows:

“The Falkland Islands support globally important populations of seabirds and coastal birds, underscoring their value for international conservation efforts.” (lines 9-10).

2. L 19-20. Insert “birds of different” before “breeding status”

The sentence was removed as the abstract was updated after the analysis was run again.

3. L 21-23. This conclusion is not logical or compelling as stated. The sentence first states that kelp beds occur “all over the archipelago” and are thus perhaps far from limiting in availability, but then warns that any changes in density of kelp beds are “expected to negatively impact the species”. To draw such a conclusion, some quantification of the area of this habitat available vs. the area of this habitat occupied by the current steamer duck population is needed.

Moreover, what do you mean by kelp “density”? Fraction of total area occupied by kelp habitats? Biomass of kelp per unit area of habitat? This qualitative conjecture could have been stated anecdotally without conducting the study, so please back it up with specific analyses.

Thank you for this feedback, we agree with your point that we did not adequately test the relationship with kelp. Now, we have included both absolute area of kelp (in m2) and proportion of kelp within both the 50% and 95% ranges of each individual. We found that an increase in absolute kelp area leads to an increase in home range size and decrease in core range size. We also found that time spent foraging increased in the home range when the absolute area of kelp increased. The opposite pattern was found for core (50%) range. We thus hypothesise that the steamer ducks will have to adjust their home and core range sizes and change their activity budget, according to the amount of kelp area and distance to kelp. To maintain the same time spent foraging, they would rest less to compensate for the increased distance they would have to travel. The sentence now reads as follows:

“ Kelp beds are present in coastal waters all over the archipelago and, consequently, likely influence the distribution and density of Steamer ducks. Therefore, any changes in their absolute area are expected to negatively impact the species.” (lines 20-23).

4. In general, the Abstract should contain more specific results. What were the distances traveled and home and core range sizes among birds of different pair status (or at least the ranges observed). What were percentages of time spent in different behaviors? What specific variables led you to your final statement that changes in the density of kelp beds would negatively the species? I suggest substantially reducing the first half of the Abstract which is all introductory, and then providing some specific, quantitative findings. The Abstract is important for people who will not read the paper, so you need to provide more substance.

Thank you for your suggestions. We reduced the introductory part, provided the main outputs from the different models, and we hope the updated version aligns more with what you were expecting to find in an abstract.

5. L 32-33. The passage should read “… invasive species, climate change, and inaccessibility often …”

Thank you for the correction. We edited as suggested. The sentence now reads as follows:

“Additional constraints to endemic species success, such as competition with invasive species and climate change (6,7) often result in data limitations hindering their conservation.

6. L 34-40. This paragraph adds little that ecologists reading the paper do not already know. I suggest deleting the paragraph.

Thank you for the suggestion. This paragraph was indeed deleted to allow more room for additional information on the coastal habitats.

7. L 41-44. The first sentence of this paragraph can be deleted with no loss of information important to this paper. Eliminate summaries of general knowledge and get to the substance of specific questions you will address in this study.

Thank you for the suggestion. This sentence was indeed deleted.

8. L 47. Explain what you mean by “passive defence”

Thank you for the comment. We added two behaviours the steamer ducks use as passive defence which are patrolling and calling. The sentence was removed as a consequence of reshaping the introduction, however.

9. L 49-50. To be more concise, I suggest ‘Territories of marine waterfowl encompass both marine and terrestrial habitats.”

Thank you for the suggestion. The sentence was indeed replaced as is (lines 39-40).

10. L 53. I strongly suggest that you avoid acronyms unless the terms are really ponderous. Busy scientists often do not have time to read a paper in one sitting within the same few days, and may skip around in a paper to find particular information. Consequently, it is quite annoying have to search back through a paper to find what a particular acronym stands for.

11. L 57. Replace “apparent” by “apparently”. The pairbonds are not apparent, they are apparently long-term.

Thank you for the correction. The sentence now reads as follows:

“Individuals of this genus form apparently long-term pair-bonds (3), guarding well-defended territories year-round (6), with incubation conducted solely by females (7) and the territory revolving around her (8).” (lines 48-50).

12. L 60. Replace “animal” by “bird and mammal” if that’s what you mean. I suspect there are endemic insects, arachnids, or other invertebrates that you have not considered in this general statement about animals.

Thank you for the suggestion. The sentence now reads as follows:

“The Falkland Steamer Duck, one of only two endemic bird and mammal species on the Falkland Islands, along with Cobb’s wren (Troglodytes cobbi, is ubiquitous along the coastlines.” (lines 50-52).

13. L 63. I suggest deleting “interfacing” as unnecessary, and replacing “comprise” by “include”

Thank you for the suggestion. The introduction was partly reshaped to describe better the importance of both the inland and at sea habitat, so this has been removed.

14. L 69. Do you mean “these factors”, referring to sex, breeding status, and location mentioned in the preceding sentence? Why would you expect these factors to result in differences in these activities? You have not commented on potential reasons in preceding text, but such expectations could provide you with more specific predictions (and related conclusions) based on ecological processes of interest to a general readership.

Thank you for your feedback. We have moved this section, and expanded on it to clarify, to the paragraph detailing the aims (lines 71-85).

15. L 83-84. How far inland do the ducks nest?

Thank you for this question. We have only one reference to this question, which has been added in as follows:

“On the Falkland Islands, on the south bank of Stanley Harbour, one pair of Falkland Steamer Ducks had a nest around 0.8km away from the shore, on the north side of the harbour (26), comprising a high proportion of shrub vegetation and ferns.” (lines 58-61).

16. L 117. Please state clearly whether “daily distance traveled” was along the trajectory of movement rather than the maximum linear distance moved between the beginning and end of the day.

Thank you for your comment. The sentence now reads as follows:

“Movement tracks were then interpolated at 2 min intervals using the adehabitatLT package (36) and daily distances travelled (km) were calculated along the trajectory of movement.” (lines 130-132).

17. L 118-119. For the many readers who will not know, please explain briefly what the “random bridge kernel method” does.

Thank you for your suggestion. The following sentences should explain better what the biased-random-bridge kernel method does:

“We defined home range based on two levels of the Utilisation Distribution (UD) obtained via the biased-random bridge kernel method (39). This method produces an occurrence distribution, rather than a true extrapolated home range, and represents a version of Brownian bridges (42) using the movement-based kernel density estimation. We estimated the wider home range (UD95) and core range (UD50), representing the area in which the individual can be found 95 % and 50 % of the tracked time.” (lines 138-143).

18. L 120. What does “UD” stand for? An intuitive variable name is desirable.

Thank you for the correction. UD is frequently abbreviated as such but is now better described in the previously mentioned paragraph:

“. We estimated the wider home range (UD95) and core range (UD50), representing the area in which the individual can be found 95 % and 50 % of the tracked time.” (lines 141-143).

19. L 123. Again, please explain in one or two sentences how “Hidden Markov Models” work.

Thak you for the comment. The following sentences were added:

“From the GPS locations of each individual, three ecologically relevant statuses (resting, foraging and travelling) were determined using Hidden Markov Models. These models rely on an observable set of data (here the tracking data for each individual), to infer a non-observable state dependent on the distance and angle between subsequent points (step length, and turning angle) (43). Using the moveHMM package (44), we defined the form of each state, which were then used to estimate the proportion (%) of time spent in each states.” (lines 155-163).

20. L 124. Delete “in each of three ecologically relevant states (i.e.”. Based on GPS readings every 2 min, please explain how you discriminated resting vs. foraging in one place, and moving among nearby foraging patches vs. commuting.

Thank you for the comment. The suggested phrase was indeed deleted, and the following sentences were added instead:

“Resting is depicted as having the lowest step length and low turning angles. Commuting is best described by rapid movements, that is to say long step length and low turning angles. Whereas sharp turning angles and lower step lengths indicating more tortuous movements, indicative of exploration, represent foraging” (lines 160-163).

21. L 148. Two days seems inadequate for defining an individual’s home range. Did you examine the data to see after how many days the home ranges mostly stabilized? Some standards for adequate sample size (in days) seem important to unbiased estimates of true home range size.

Thank you for pointing this issue out, we agree that confirmation is necessary to know which individuals to use. We found that even with two days of data for some individuals we were able to calculate a stable home range. We did however remove some individuals from the analysis after conducting preliminary exploration.

We used the variograms from the workflow described by Calabrese et al 2016. Any individual where semi-variance was reached and remained as an asymptote were kept for the home and core range analysis. The following sentences were added:

“Firstly, to identify resident individuals for whom we had sufficient data to calculate the home range, we produced variograms using the ‘variogram’ function from the ctmm package. Individuals were considered resident when the variograms approached an asymptote (39). Any individuals that did not reach or maintain an asymptote were not considered for home and core range analysis.” (lines 134-138).

22. Figure 2. Panels A, B, D, and E are nice depictions of the ducks’ movements. However, the green areas that supposedly show foraging areas in panels C and F are effectively impossible to see, even when scrutinized with a magnifying glass. A reader with red-green color blindness would have no chance of discriminating what is shown in this figure. I suggest that panels C and F be moved into a separate figure, enlarged, and alternative colors selects (perhaps yellow for foraging).

What is represented by the areas in beige?

Thank you and apologies for the for the oversight. The beige represents kelp beds. Figure 2 and its caption were changed accordingly. The caption now reads as follows:

“Fig. 2 Movement tracks of birds tagged on Stanley (A) and Bleaker Island (C), with their associated home and core ranges (respectively B and D). Home and core ranges with sufficient data were coloured in black (males) and red (females) while nominal home and core ranges with insufficient data were coloured in light grey (males) and orange (females). Kelp beds were represented in beige.” (lines 193-196).

23. Table 1. Please state clearly in the caption whether the distances traveled were along the trajectory of movement or were the linear distance between points at the start and end of the day. Please also specify “km/day” (not just km) as the units in the table heading. How did you standardize the core and home range sizes among individuals that were tracked for a little as 2 days to as much as 43 days (see L 148). Ranges of the birds are likely to differ substantially between such short and long tracking periods. Without better explanation of how you standardized and calculated these values, it’s not clear that results for different groups can be directly compared.

Thank you for your detailed feedback. The table was changed accordingly (line 203).

As now mentioned in the methods, only individuals with enough data were kept (see comment 21). This implies that enough data were collected to map reliably the home and core ranges. Additionally, models were weighted with track duration when needed:

“For all models, track duration was added as a weight when relevant and fit was assessed by simulating residuals in the DHARMa package.” (lines 169-170).

24. Tables 1 and 2 are both labeled as Table 1. “Dependent” is misspelled in what should be Table 2.

Thank you for highlighting those issues. They have been corrected according, and table 2 now sits in the supplementary information so as not to disrupt the flow of the paper (line 195 and S3 Table).

25. Table 2. Here and elsewhere in the paper, I suggest replacing “location” with “study site” to clarify what you mean (if in fact that’s what you mean). In this paper, you have locations every 2 min via the GPS, but two different study sites.

I also suggest “dist to kelp” and “dist to structure”, which are not much longer and clarify what the variable represents.

Thank you for these suggestions. Table 2 was changed accordingly and “location” replaced by “study site” throughout the manuscript.

26. Table 2. The second and third variables for Distance per day are exactly the same, yet you assign them different values of AIC and ΔAIC.

---

## [Editor Report · Decision Letter 1]

11 Sep 2025

Home range and activity budget in the Falkland Steamer Duck (Tachyeres brachypterus)

PONE-D-25-23682R1

Dear Alix,

I have reviewed the significant changes you have made to the prior version of the manuscript in response to the two reviews and I am pleased to inform you that I judge that your manuscript has been revised satisfactorily and is suitable for publication. It will be formally accepted for publication once it meets any outstanding technical requirements identified by the Editorial Office. Thank you for considering PLOS ONE and for preparing this contribution that improves knowledge of this understudied waterfowl species. 

The sequence of events to follow, leading to the publication of your paper are as follows:

Within one week, you’ll receive an e-mail detailing any required amendments. When these have been addressed, you’ll receive a formal acceptance letter and your manuscript will be scheduled for publication.

Kind regards,

Lee W Cooper, Ph.D.

Section Editor

PLOS ONE

---

## [Editor Report · Acceptance letter]

PONE-D-25-23682R1

PLOS ONE

Dear Dr. Kristiansen,

I'm pleased to inform you that your manuscript has been deemed suitable for publication in PLOS ONE. Congratulations! Your manuscript is now being handed over to our production team.

Kind regards,

on behalf of

Dr. Lee W Cooper

Section Editor

PLOS ONE